# Ibuprofen Favors Binding of Amyloid-β Peptide to Its Depot, Serum Albumin

**DOI:** 10.3390/ijms23116168

**Published:** 2022-05-31

**Authors:** Ekaterina A. Litus, Alexey S. Kazakov, Evgenia I. Deryusheva, Ekaterina L. Nemashkalova, Marina P. Shevelyova, Andrey V. Machulin, Aliya A. Nazipova, Maria E. Permyakova, Vladimir N. Uversky, Sergei E. Permyakov

**Affiliations:** 1Institute for Biological Instrumentation, Pushchino Scientific Center for Biological Research of the Russian Academy of Sciences, 142290 Pushchino, Russia; fenixfly@yandex.ru (A.S.K.); janed1986@ya.ru (E.I.D.); elnemashkalova@gmail.com (E.L.N.); marina.shevelyova@gmail.com (M.P.S.); alija-alex@rambler.ru (A.A.N.); mperm1977@gmail.com (M.E.P.); 2Skryabin Institute of Biochemistry and Physiology of Microorganisms, Pushchino Scientific Center for Biological Research of the Russian Academy of Sciences, 142290 Pushchino, Russia; and.machul@gmail.com; 3Department of Molecular Medicine, Byrd Alzheimer’s Research Institute, Morsani College of Medicine, University of South Florida, Tampa, FL 33612, USA; vuversky@usf.edu

**Keywords:** Alzheimer’s disease, amyloid-β peptide, human serum albumin, ibuprofen, surface plasmon resonance, Aβ fibrillation, electron microscopy, molecular docking

## Abstract

The deposition of amyloid-β peptide (Aβ) in the brain is a critical event in the progression of Alzheimer’s disease (AD). This Aβ deposition could be prevented by directed enhancement of Aβ binding to its natural depot, human serum albumin (HSA). Previously, we revealed that specific endogenous ligands of HSA improve its affinity to monomeric Aβ. We show here that an exogenous HSA ligand, ibuprofen (IBU), exerts the analogous effect. Plasmon resonance spectroscopy data evidence that a therapeutic IBU level increases HSA affinity to monomeric Aβ_40_/Aβ_42_ by a factor of 3–5. Using thioflavin T fluorescence assay and transmission electron microcopy, we show that IBU favors the suppression of Aβ_40_ fibrillation by HSA. Molecular docking data indicate partial overlap between the IBU/Aβ_40_-binding sites of HSA. The revealed enhancement of the HSA–Aβ interaction by IBU and the strengthened inhibition of Aβ fibrillation by HSA in the presence of IBU could contribute to the neuroprotective effects of the latter, previously observed in mouse and human studies of AD.

## 1. Introduction

About 70–80% of all dementia cases (over 55 million people worldwide, according to World Health Organization, WHO) correspond to Alzheimer’s disease (AD), which represents a progressive neurodegenerative disorder leading to brain atrophy and death (reviewed in ref. [1,2]). Due to increasing life expectancy in developed countries, AD is a global threat, lacking effective curative or preventive means despite the enormous efforts of the scientific community and multibillion research investments. The key histopathological hallmarks of AD are the accumulation of harmful extracellular amyloid β peptide (Aβ) plaques and neurofibrillary tangles of hyperphosphorylated tau protein in the hippocampus and neocortex [2,3,4]. The Aβ deposition begins 2–3 decades prior to clinical manifestation of AD and precedes the development of tau pathology, which better correlates with the appearance of clinical symptoms [2,3]. Meanwhile, the definite crosstalk between Aβ and tau protein observed in immunotherapy of AD points out that both these pathways should be targeted in AD therapy [3]. Indeed, the anti-Aβ monoclonal antibody aducanumab recently became a first targeted treatment approved by the Food and Drug Administration for AD, whereas two other antibodies targeting Aβ await their accelerated approval [2,5]. Therefore, the anti-Aβ therapy proved to be a viable tactic for the prevention of AD.

The Aβ is a 37–49-residue-long peptide enzymatically cleaved from the amyloid precursor protein, with the two major forms being: the 40-residue Aβ_40_ and the 42-residue Aβ_42_ [2,6]. Multimerization of monomeric Aβ leads to the formation of soluble toxic oligomers, protofibrils, and insoluble mature fibrils, which assemble into amyloid plaques [6]. Therefore, the Aβ therapeutics in clinical trials pursue avoidance of Aβ aggregation, aside from reduction of Aβ production and enhancement of Aβ clearance [3]. Notably, some of the plasma and cerebrospinal fluid (CSF) proteins prevent Aβ polymerization via binding to Aβ, with the highest contribution (over 60%) being from the serum albumin [7]. In fact, the most abundant blood plasma and CSF protein (60–80% of the total protein [8]), human serum albumin (HSA), binds about 89% of the total Aβ in plasma [9]. The much lower HSA level in CSF compared to that in the blood serum (3 μM versus ca. 645 μM [10], respectively) likely explains the fact that the Aβ plaques accumulate in the central nervous system but not in the peripheral tissues, where the abundant HSA efficiently suppresses Aβ multimerization [7]. The protective effect of plasma HSA is further supported by its antioxidant activity due to the C34 thiol and detoxification activity owing to the HSA’s ability to bind a wide range of substances, including endogenous toxins [8]. Meanwhile, even the relatively low HSA level observed in CSF inhibits Aβ fibrillation in vitro [11], thereby indicating the therapeutic potential of HSA.

The Aβ-buffering role of HSA could be exploited for AD therapy by directed improvement of HSA’s affinity to Aβ, as exemplified by the addition of certain low-molecular-weight ligands of HSA: arachidonic/linoleic acid or serotonin (the maximal effect is a 17-fold increase in HSA’s affinity to Aβ [12,13]). Other HSA ligands typically act in the opposite direction, as shown for tolbutamide [7], cholesterol, palmitic acid, and warfarin [14]. The more radical approach to therapeutic use of HSA is Aβ removal via replacement of endogenous albumin with therapeutic HSA through a plasma exchange [15]. The respective clinical trials evidence slowdown of cognitive and functional decline in AD [16,17,18]. Finally, direct delivery of exogenous HSA into the murine brain via intracerebroventricular administration gave positive results [19].

In the present work, we show that therapeutic levels of one more HSA ligand, ibuprofen (2-[4-(2-methylpropyl)phenyl]propanoic acid, IBU), notably improve HSA affinity to monomeric Aβ, leading to the enhanced suppression of the Aβ fibrillation by HSA in vitro. IBU is an over-the-counter, non-steroidal, anti-inflammatory drug from the Essential Medicines List of the WHO, widely used for relief of pain, fever, and inflammation (reviewed in [20]). Its therapeutic indications include rheumatoid- and osteo-arthritis, cystic fibrosis, orthostatic hypotension, dental pain, dysmenorrhea, fever, and headache [21]. IBU exerts multifactorial action on the different pathways involved in acute and chronic inflammation, including inhibition of the prostaglandin synthesis via inhibition of the cyclooxygenases COX-1 and COX-2, suppression of the leucocyte functions, modulation of the nitric oxide and cytokine production, etc. [20]. The water solubility of ibuprofen at pH values above 7 (pKa = 5.3) reaches 0.1–1 M [22,23]. At therapeutic levels, 99% of IBU is bound to blood plasma proteins [24]. HSA binds single IBU molecules with an equilibrium dissociation constant, *K_d_*, of 0.4 µM and 6–7 IBU molecules with a *K_d_* value of 51 µM [25]. The primary IBU-binding site of HSA is located in the center of the drug site 2 of the subdomain IIIA (sites FA3-FA4), whilst the secondary site is located at the interface between subdomains IIA and IIB in a cleft that overlaps with the site FA6 [26]. IBU binding stabilizes the tertiary structure of HSA but does not affect its secondary structure [27].

Numerous studies evidence various neuroprotective effects of IBU intake in mouse models of AD, including suppression of the inflammatory processes, reduction in the Aβ sprand tau protein depositions, and lowering of the cognitive and memory deficits [28,29,30,31,32,33,34,35]. Similarly, retrospective human epidemiological studies revealed that long-term use of nonsteroidal anti-inflammatory drugs, including IBU, is protective against AD [36,37,38]. The IBU-induced Aβ trapping by HSA shown here in vitro points out the existence of an additional molecular mechanism behind these observations, which likely complements the other positive effects of IBU, such as suppression of chronic inflammation and Aβ42 level, scavenging of free radicals, and γ-secretase modulation [20,39,40].

## 2. Results and Discussions

### 2.1. Modulation of HSA Affinity to Monomeric Aβ by IBU

The interaction between fatty-acid-free HSA and Aβ at 25 °C was studied by surface plasmon resonance (SPR) spectroscopy mainly as described earlier [12,13]. The recombinant human Aβ_40_/Aβ_42_ or Flemish variant of Aβ_40_ [41], Aβ_40_(A21G), was immobilized on the surface of the SPR sensor chip by amine coupling. To ensure the monomeric state of Aβ, the non-covalently bound Aβ molecules were thoroughly washed from the chip surface until stabilization of the SPR signal using several water solutions: 2% sodium dodecyl sulfate (SDS), 100 mM HCl, 20 mM ethylenediaminetetraacetic acid (EDTA) pH 8.0 with 1% SDS. The defatted HSA sample extracted from blood under non-denaturing conditions [42] was used as an analyte. Briefly, 0.5–8 µM HSA in the physiologically relevant running buffer (20 mM Tris-HCl, 140 mM NaCl, 4.9 mM KCl, 1 mM MgCl_2_, 2.5 mM CaCl_2_, pH 7.4) was passed over the chip surface for 300 s, followed by flushing of the chip with the running buffer. The measurements were performed either in the absence or presence of 100 µM IBU, which corresponds to therapeutic IBU level in plasma [43,44] and is sufficient for HSA saturation with IBU [25]. The resulting SPR data exhibited a characteristic concentration-dependent association–dissociation pattern and were well described by the heterogeneous ligand model (1) (Figure 1) with the kinetic and equilibrium association/dissociation constants shown in Table 1.

Unexpectedly, *K_d_* for HSA–Aβ_40_/Aβ_42_ complexes are 2–3 orders of magnitude lower compared to those reported earlier under identical solution conditions [13]. This inconsistency can be resolved considering that we washed the non-covalently bound Aβ molecules from the chip surface using two additional solutions, aside from the 2% SDS solution used earlier [13], thereby ensuring a more monomeric state of Aβ. Notably, despite the differences in the absolute values of the *K_d_* estimates obtained here and in the previous works [12,13], the general regularities remain the same: HSA’s affinities to Aβ_40_ and Aβ_42_ are nearly equal, whereas the A21G mutation increases HSA’s affinity to Aβ_40_ by a factor of 2.5–10 [12] or 1.5–4 (Table 1).

Addition of the 100 µM IBU notably favors the HSA–Aβ_40_/Aβ_42_ interaction: the *K_d_* values are decreased by a factor of 3–5 (Table 1). The effect is less noticeable in the case of Aβ_40_(A21G): the *K_d_* values are lowered by 60–70%. In all cases, the *K_d_* decline induced by IBU is mainly due to the slowdown in the HSA–Aβ complex dissociation, which is manifested as a flattening of the dissociation phase of the SPR curve (Figure 1).

### 2.2. Influence of IBU on Suppression of Aβ Fibrillation by HSA

The improved Aβ trapping by HSA in the presence of IBU (Table 1) is expected to decrease the free Aβ concentration, leading to the slowdown of the Aβ fibrillation. HSA per se is able to interact with monomeric and multimeric forms of Aβ, thereby interfering with the Aβ fibrillation [7,9,11,14,45,46]. Furthermore, some of the HSA ligands are shown to affect the suppression of Aβ fibrillation by HSA [7,14]. To explore the influence of IBU on this process, we used the thioflavin T (ThT) fluorescence assay [47,48]. Kinetics of the fibrillation process for 20 µM Aβ_40_/Aβ_40_(A21G) in the absence/presence of 2–10 µM HSA (corresponds to HSA level in cerebrospinal fluid of 3 µM [10]) or/and 20 µM IBU (close to therapeutic IBU level [43,44]) at 30 °C was followed by fluorescence of 10 µM ThT for 190 h (Figure 2 and Figure 3).

In accord with the literature data for Aβ_40_ [11], addition of the 5–10 µM HSA drastically decreases the final ThT fluorescence intensity, indicating the prevention of Aβ_40_/Aβ_40_(A21G) fibrillation (Figure 2A,C). Although 20 µM IBU alone does not induce statistically significant suppression of the fibrillation process (Figure 2B,D and Figure 3), the addition of IBU strengthens the suppression of Aβ_40_ fibrillation by 2 µM HSA (Figure 3A). Meanwhile, the fibrillation of Aβ_40_(A21G) is efficiently prevented by 2 µM HSA regardless of IBU’s presence (Figure 2C and Figure 3B).

Since the kinetics of the Aβ_40_ fibrillation (Figure 3A) is adequately described by the Boltzmann sigmoid function (Equation (2)), the fibrillation process was quantitated by the values of lag time (*t_lag_*, Figure 4A) and apparent rate constant (*k_app_*, Figure 4B). The addition of 20 µM IBU did not affect the *k_app_* value but shortened the lag phase by a factor of 2, pointing out the presence of the IBU interaction with Aβ_40_, in line with the previous experimental and theoretical studies [49,50,51,52]. Meantime, 2 µM HSA prolonged the lag phase of Aβ_40_ fibrillation by ~60% regardless of IBU presence, with a decline in the *t_lag_* value by 21% in response to IBU. The combination of HSA and IBU decreased the *k_app_* value by 17%.

It should be noted that the efficient suppression of 25–50 µM Aβ_40_/Aβ_42_ fibrillation was previously reported at an IBU concentration as low as 10 µM [49]. The notable discrepancy with our data could be due to the use in the previous study [49] of synthetic Aβ samples, which exhibit altered properties (lowered propensity to fibrillation and neurotoxicity [53]), and to differences in Aβ pretreatment.

Overall, the ThT fluorescence assay reveals that 20 µM IBU favors the suppression of fibrillation of 20 µM Aβ_40_ by 2 µM HSA along with the minor opposite effects: shortening of the lag phase and decrease in the apparent rate constant of the process.

### 2.3. The Suppression of Aβ_40_ Fibrillation by HSA/IBU Studied by Transmission Electron Microscopy

To explore the structural features of the fibrils grown in the experiment shown in Figure 3A, we studied the resulting fibers using negative staining transmission electron microscopy (TEM) (Figure 5). The 20 μM Aβ_40_ sample reveals dense clusters of the intertwined mature fibrils up to microns long (Figure 5A). The fibrillation in the presence of 20 μM IBU gives rise to drastically less fuzzy fibrils, which indicates suppression of the fibrillation process (Figure 5B). This result correlates with the somewhat lowered ThT fluorescence intensity (Figure 2B and Figure 3A) and altered *t_lag_* value (Figure 4A) in the presence of IBU and the literature data [49]. Meanwhile, 2 μM HSA prevented the formation of long fibers, and the fibrillation process was notably suppressed (Figure 5C). Furthermore, the combined application of 2 μM HSA and 20 μM IBU in the course of Aβ_40_ fibrillation caused the almost complete disappearance of the mature fibers (Figure 5D). Taken together, the TEM data evidence the suppressive action of HSA and its combination with IBU on the Aβ_40_ fibrillation, in accord with the ThT fluorescence assay data (Figure 3A). Since the effect of IBU is opposite to that exerted by other HSA ligands, including tolbutamide [7], cholesterol, palmitic acid, and warfarin [14], one may suggest that the IBU-binding site of HSA differs from the site(s) occupied by these molecules. The latter conclusion is in line with the data on the relative location of IBU/warfarin-binding sites of HSA [26].

### 2.4. Structural Modeling of HSA–Aβ_40_/IBU Complexes

The previous experimental and molecular dynamics study [45] has shown that the HSA groove between domains I and III is the most probable binding site for the Aβ_40_/Aβ_42_ monomer. The binding process was accompanied by the conversion of Aβ structure from the random coil into α-helical structure, while HSA did not show noticeable structural changes [45]. Therefore, we predicted the location of Aβ_40_-binding sites of HSA using ClusPro docking service [54] (Figure 6A), based upon crystal structure of HSA (PDB code 1AO6) and the NMR structure of the partially folded α-helical Aβ_40_ (PDB code 2LFM, Figure 6B). In accord with the previous report [45], the primary Aβ_40_-binding site of HSA was predicted to be located in the groove between domains I and III (Figure 6A). Since residue A21 of Aβ_40_ contacts with HSA molecule in 7 of the 25 model complex structures, the A21G substitution is expected to affect the HSA-Aβ_40_ interaction, in line with the SPR data (Table 1). Furthermore, a secondary Aβ_40_-binding site located in the domain II of HSA was predicted (Figure 6A). This site is in the close vicinity to the previously reported IBU-binding sites of HSA (Figure 6A,C,D) [26]. Furthermore, one of them intersects with the secondary site of Aβ_40_ (residues A213 and K351 of domain II and residues L481-V482 of domain III; Figure 6D). The partial overlap between IBU/Aβ_40_-binding sites of HSA could interfere with the simultaneous binding of IBU and Aβ_40_ to HSA. Meanwhile, the IBU-induced increase in HSA affinity to Aβ_40_ (Table 1) rules out this possibility. Furthermore, this effect could be attributed to the direct IBU interaction with Aβ_40_, as reported earlier [51,52].

Curiously, Figure 7 shows that HSA residues comprising the IBU binding sites possess prominent difference in the intrinsic disorder propensity, with residues of the primary binding site being noticeably more flexible that residues of the secondary binding site. This can be illustrated by the intrinsic disorder propensity evaluated for these residues by PONDR^®^ VSL2, which is one of the more accurate per-residue disorder predictors. In fact, the disorder propensities of the residues forming primary site were distributed more uniformly and ranged from 0.2234 to 0.5046, with the mean disorder score of this site being 0.347 ± 0.084. On the other hand, the intrinsic disorder propensities of the secondary site residues were more diversified, with their disorder scores ranging from 0.0796 to 0.5363. As a result, the mean disorder score of this site was 0.26 ± 0.17. Furthermore, residues involved in the overlapping IBU/Aβ_40_-binding sites were characterized by the most uniform distribution of their intrinsic disorder predispositions, with Ala213, Lys351, Leu481, and Val482 showing disorder scores of 0.4271, 0.2113, 0.2888, and 0.2712 (mean disorder score 0.300 ± 0.091). These observations suggest that the efficiency of the IBU and Aβ_40_ binding by HSA can be linked to the local intrinsic flexibility of this protein (i.e., flexibility encoded in the amino acid sequence and not in the 3D structure and reflected in the local peculiarities of the per-residue intrinsic disorder profile). Interestingly, stronger IBU binding to the primary site can be due to the higher overall flexibility of this site that can better adjust to the ligand at binding.

## 3. Materials and Methods

### 3.1. Materials

Fatty-acid-free has prepared under non-denaturing conditions [42] was purchased from Merck (#126654). Ubiquitin carboxyl-terminal hydrolase 2, catalytic core, (Usp2-cc) was prepared mainly as described in ref. [56]. IBU was purchased from Acros Organics. Ultra-grade Tris and 2-mercaptoethanol (2-ME) were from Amresco^®^ LLC (Vienna, Austria). Urea, imidazole, sodium chloride, potassium chloride, sodium hydroxide, SDS, and glycerol were purchased from Panreac AppliChem (Darmstadt, Germany). Calcium chloride and magnesium chloride were from Fluka (Charlotte, NC, USA). EDTA, ThT, and polyethylene glycol sorbitan monolaurate (TWEEN^®^) 20 were from Sigma-Aldrich (St. Louis, MO, USA). Ethanolamine and Profinity^TM^ IMAC resin were bought from Bio-Rad Laboratories (Hercules, USA). Hydrochloric acid was from Sigma Tec LLC (Moscow, Russia). Dimethyl sulfoxide (DMSO) was from Helicon (Moscow, Russia). Trifluoroacetic acid (TFA) was purchased from Fisher Scientific (Madrid, Spain). Sodium azide was from Dia-M (Moscow, Russia).

Protein concentrations were measured spectrophotometrically using molar extinction coefficients at 280 nm calculated according to ref. [57]: 34,445 M^−1^cm^−1^ for HSA and 1490 M^−1^cm^−1^ for Aβ_40_/Aβ_40_(A21G)/Aβ_42_ at pH 7.4–8.0.

Stock solution of ThT (0.8 mg/mL) was prepared in distilled, deionized water. The ThT concentration was measured spectrophotometrically using the molar extinction coefficient at 412 nm of 36,000 M^−1^cm^−1^ [58].

### 3.2. Preparation of Recombinant Human Aβ Samples

Human Aβ_40_/Aβ_42_/Aβ_40_(A21G) were expressed in *E. coli* and purified as described earlier [12,13] with the following modifications. The cells were disintegrated by wet grinding using a Retsch^®^ Mixer Mill MM 400 (glass beads with diameter of 0.5–0.75 mm, grinding time 1 min, 7 cycles). The supernatant was loaded onto a Profinity^TM^ IMAC resin column (5 mL) equilibrated with buffer A (50 mM Tris-HCl, 8 M urea, 5 mM 2-ME, pH 8.2). The column was shaken at 10–15 rpm for 60 min at room temperature using a rotator, followed by sequential washing of the column with 50 mL of buffer A with 5 mM imidazole, 20% glycerol, and 0.5% TWEEN^®^ 20; 50 mL of buffer A with 5 mM imidazole and 20% glycerol; and 50 mL of buffer A with 5 mM imidazole. The 6× His–ubiquitin–Aβ fusion protein was eluted from the column with a linear gradient of 10–300 mM imidazole (60 mL). Aβ was excised from the fusion protein by the Usp2-cc treatment. To remove the His-tagged ubiquitin and Usp2-cc, the hydrolysate was loaded onto a Profinity^TM^ IMAC resin column equilibrated with buffer A, the column was shaken at 10–15 rpm for 60 min at room temperature. The unbound fraction containing Aβ was purified by high-performance liquid chromatography using a Phenomenex^®^ Jupiter C18 column. Precise chain cleavage by Usp2-cc was confirmed by electrospray ionization mass spectrometry (Shimadzu LCMS-2010EV). The purified Aβ samples were freeze-dried and stored at −70 °C.

### 3.3. Preparation of Aβ Samples for SPR Experiments

The human Aβ_40_/Aβ_42_/Aβ_40_(A21G) samples were pretreated prior to SPR studies mainly as described in ref. [59]. The freeze-dried Aβ samples were dissolved in neat TFA at a concentration of 0.5–1 mg/mL, followed by sonication for 30 s and TFA evaporation using an Eppendorf Concentrator plus. The dried Aβ samples were dissolved in DMSO at a concentration of 2 mg/mL and stored at −20 °C.

### 3.4. Surface Plasmon Resonance Studies

SPR measurements of HSA affinity to monomeric Aβ samples at 25 °C were performed using a Bio-Rad ProteOn™ XPR36 protein interaction array system similarly to the procedure described in refs. [12,13]. The pretreated ligand (0.05 mg/mL Aβ_40_/Aβ_40_(A21G)/Aβ_42_ in 10 mM sodium acetate, pH 4.5 buffer) was immobilized on a ProteOn™ GLH sensor chip surface by amine coupling up to 6000–9000 RUs. The rest of the activated amine groups on the chip surface were blocked by 1 M ethanolamine solution. The non-covalently bound ligand molecules were sequentially washed from the chip surface until stabilization of the SPR signal using the following water solutions: 2% SDS, 100 mM HCl, 20 mM EDTA pH 8.0 with 1% SDS. The analyte (0.5–8 μM HSA) in the running buffer (20 mM Tris-HCl, 140 mM NaCl, 4.9 mM KCl, 1 mM MgCl_2_, 2.5 mM CaCl_2_, pH 7.4) with/without 100 µM IBU was passed over the chip at a rate of 30 μL/min for 300 s, followed by flushing the chip with the running buffer for 2400 s. The sensor chip surface was regenerated by passage of a water solution of 0.5% SDS. The double-referenced SPR sensograms were analyzed using the heterogeneous ligand model (Equation (1)), assuming presence of two populations of a ligand (L1 and L2) that bind an analyte molecule (A):(1)$$\;\begin{array}{ccc} & k_{a1} & \\ L_{1}+\;A & ⇌ & L_{1}A\\ & \begin{array}{c} k_{d1}\\ K_{d1} \end{array} & \end{array}\;\;\;\;\;\begin{array}{ccc} & k_{a2} & \\ L_{2}+\;A & ⇌ & L_{2}A\\ & \begin{array}{c} k_{d2}\\ K_{d2} \end{array} & \end{array}$$
where *k_a_* and *k_d_* refer to kinetic association and dissociation constants, respectively; *K_d_*_1_ and *K_d_*_2_ are equilibrium dissociation constants. The equilibrium and kinetic dissociation/association constants were evaluated for each analyte concentration using Bio-Rad ProteOn Manager™ v.3.1 software (Hercules, CA, USA), followed by averaging of the resulting values (n = 3–4; standard deviations are indicated).

### 3.5. Preparation of Aβ Samples for ThT Fluorescence Assay

The human Aβ_40_/Aβ_40_(A21G) samples were dissolved in 10 mM NaOH at pH ~12 (0.5 mg/mL) and then rocked gently for 72 h at 4 °C. Since Aβ lacks W residues and contains a single Y residue, Aβ concentrations were determined spectrophotometrically using the molar extinction coefficient of tyrosinate at 293 nm of 2330 M^−1^cm^−1^ [60].

### 3.6. ThT Fluorescence Assay

ThT fluorescence emission measurements were carried out using a BioTek Synergy H1 multimode microplate reader and Greiner Bio-One non-binding microplates #781906 mainly as described in ref. [11]. Briefly, 20 µM Aβ_40_/Aβ_40_(A21G) in 25 mM Tris-HCl, 140 mM NaCl, 4.9 mM KCl, 1 mM MgCl_2_, 2.5 mM CaCl_2_, pH 7.4 buffer with 0.05% NaN_3_ was incubated with 10 µM ThT in the absence/presence of 2–10 µM HSA and/or 20 µM IBU at 30 °C. ThT fluorescence was excited at 440 nm and emission at 485 nm was measured for 190 h every 30 min, with orbital shaking prior to each measurement. The kinetic fluorescence data for Aβ_40_ were described by the Boltzmann sigmoid function (Equation (2)) using OriginPro v.9.0 (OriginLab Corp.) software:(2)$$y=\frac{A_{1}-A_{2}}{1+e^{\left(t-t_{0}\right)\times k_{app}}}+A_{2}$$
where *A*_1_ and *A*_2_ are the initial and final fluorescence levels, *t*_0_ is the half-transition time and *k_app_* is the apparent rate constant of the Aβ_40_ fibrillation. The lag time, *t_lag_*, is calculated as (*t*_0_ − 2/*k*_app_) [61]. Each measurement was performed in 3–5 repetitions, and the mean *k_app_* and *t_lag_* values with standard deviations are shown.

### 3.7. Transmission Electron Microscopy

The samples after the ThT fluorescence assay (HSA concentration of 2 µM) were diluted 2-fold using 25 mM Tris-HCl, 140 mM NaCl, 4.9 mM KCl, 1 mM MgCl_2_, 2.5 mM CaCl_2_, pH 7.4 buffer. A copper 300 mesh grid coated with a formvar film (0.2%) was put on a sample drop (10 µL). After 7 min of the sample absorption, the grid was negatively stained for 2 min with UranyLess (Electron Microscopy Sciences, Hatfield, CA, USA). The excess of the staining agent was removed with a filter paper and deionized water rinse for 2 min. The samples were studied using a Tecnai G2 Spirit Bio(TWIN) transmission electron microscope (FEI Company, Czech Republic) (120 keV), equipped with a high-resolution ORIUS SC 1000B CCD camera (Gatan, Inc., Pleasanton, CA, USA).

### 3.8. Structural Modeling of Aβ_40_/IBU–HSA Complexes

The molecular modeling was based upon the structures of human Aβ_40_, HSA, and its complex with IBU (PDB [62] entries 2LFM, 1AO6, and 2BXG, respectively). In all, 25 models of the Aβ_40_–HSA complex were generated using ClusPro docking server [54]. The balanced scoring scheme was used for calculations of the interaction energies. The contact residues in the docking models of Aβ_40_–HSA complex were calculated using Python 3.3 programming language (implemented in PyCharm v.3.0.2 (Saint Petersburg, Russia) development environment), Matplotlib Python plotting library, and NumPy numerical mathematics extension. Examination of the contact residues in the HSA–IBU complex was performed using the PLIP service [55]. The numbering of the contact residues is according to the PDB entries. The tertiary structure models were drawn with molecular visualization system PyMOL v.1.6.9.0 (New York, NY, USA) [63].

### 3.9. Per-Residue Intrinsic Disorder Predisposition of HSA

The intrinsic disorder predisposition of HSA (residues 25–609 of UniProt ID: P02768) was evaluated using the web crawler RIDAO that aggregates the results from six well-known per-residue disorder predictors: PONDR^®^ VLXT [64], PONDR^®^ VL3 [65], PONDR^®^ VSL2 [66], PONDR^®^ FIT [67], IUPred2-Short, and IUPred2_Long [68,69,70,71]. Details of the local HSA disorder propensity in the IBU binding sites 1 and 2 were further characterized by PONDR^®^ VSL2, which belongs to the group of the most accurate disorder predictors as evidenced by the results of the recently conducted ‘Critical assessment of protein intrinsic disorder prediction’ (CAID) experiment, where PONDR^®^ VSL2 was recognized as predictor #3 of the 43 evaluated methods [72].

## 4. Conclusions

Epidemiological studies have reported lower rates of AD among people who had been taking IBU for chronic treatment of inflammatory conditions [36,37,73]. The reduction in brain deposition of Aβ upon treatment with IBU has been shown in several mouse AD models [28,29,30,32,33,34,35]. This effect has been ascribed to several indirect mechanisms, including downregulation of β-secretase 1 and COX-2 [28], reduction in Aβ42 production [32], inhibition of interleukin-1β and its downstream target α1-antichymotrypsin [33,34], attenuation of NADPH oxidase activation and reactive oxygen species production [35]. The in vitro data presented here suggest an additional modality, in which IBU prevents Aβ from deleterious multimerization due to synergistic action of the following factors: the improvement of Aβ trapping by HSA, enhancement of inhibitory action of HSA toward Aβ fibrillation, as well as direct prevention of Aβ fibrillation/aggregation by IBU [49,50]. One may expect both an allosteric regulation of HSA affinity to monomeric Aβ by IBU and the more complicated mechanism, in which IBU binding to Aβ monomer [51,52] alters Aβ affinity to HSA. For instance, IBU was shown to allosterically affect HSA affinity to heme [74,75] and lorazepam [76]. In the latter case, IBU binding to HSA enhanced its affinity to lorazepam.

We have previously shown that serotonin and some major plasma unsaturated fatty acids (arachidonic/linoleic acid) enhance HSA affinity toward monomeric Aβ [12,13]. The effect observed in the case of IBU (Table 1) exceeds that for the unsaturated fatty acids [12] but is lower than the effect for serotonin [13]. The HSA sites occupied by IBU (Figure 6) do not intersect with the previously predicted serotonin-binding site [13], indicating that the increase in HSA affinity to monomeric Aβ can be achieved via ligand binding to different sites. Meanwhile, other HSA ligands, such as tolbutamide [7], cholesterol, palmitic acid, and warfarin [14], exert the opposite effect, thereby demonstrating that a search of the substances favoring HSA interaction with Aβ represents a non-trivial task. This conclusion is further supported by the fact that tryptophan despite minor structural differences from serotonin and the same predicted binding pocket does not affect the HSA–Aβ_40_/Aβ_42_ equilibrium [13].

The findings presented here expand our knowledge of the potential for the directed reduction of free Aβ concentration and the suppression of its fibrillation by HSA using endogenous and exogenous HSA ligands.

## Figures and Tables

**Figure 1 ijms-23-06168-f001:**
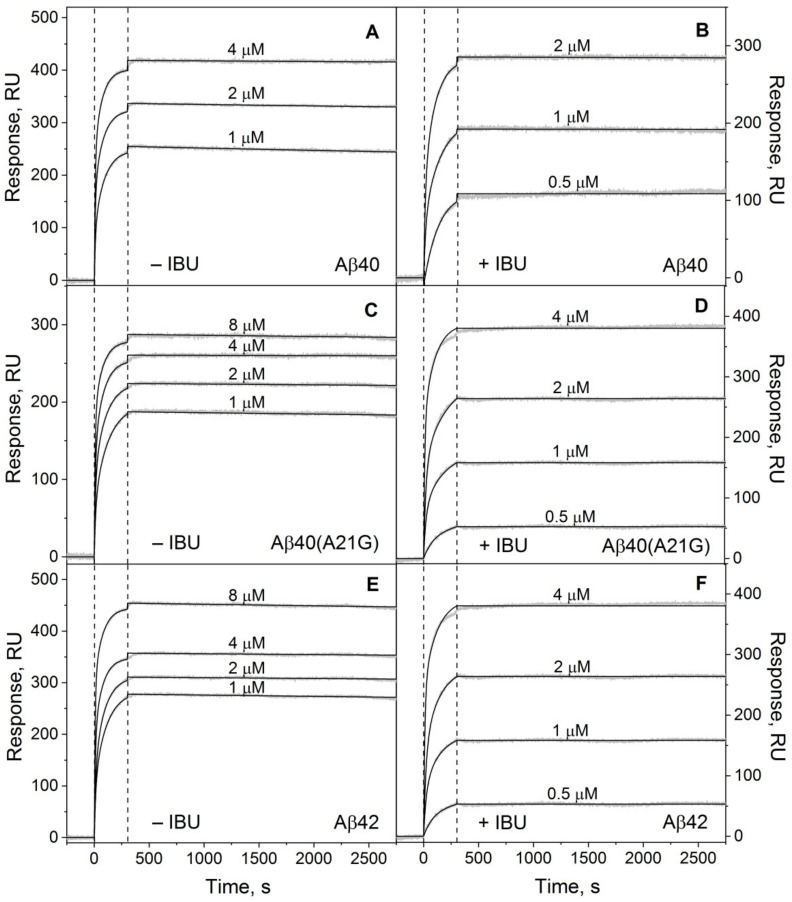
Influence of IBU on HSA’s interaction with monomeric Aβ_40_/Aβ_40_(A21G)/Aβ_42_ at 25 °C studied by SPR spectroscopy (20 mM Tris-HCl, 140 mM NaCl, 4.9 mM KCl, 1 mM MgCl_2_, 2.5 mM CaCl_2_, pH 7.4). The Aβ was immobilized on the sensor chip’s surface by amine coupling, followed by removal of the non-covalently bound Aβ molecules. HSA concentrations are indicated near the sensograms for its interaction with Aβ_40_ (**A**,**B**), Aβ_40_(A21G) (**C**,**D**), or Aβ_42_ (**E**,**F**) in the absence (**A**,**C**,**E**) or presence of 100 µM IBU (**B**,**D**,**F**). The gray curves are experimental, while the black curves are theoretical, calculated using the heterogeneous ligand model (1) (see Table 1 for the fitting parameters).

**Figure 2 ijms-23-06168-f002:**
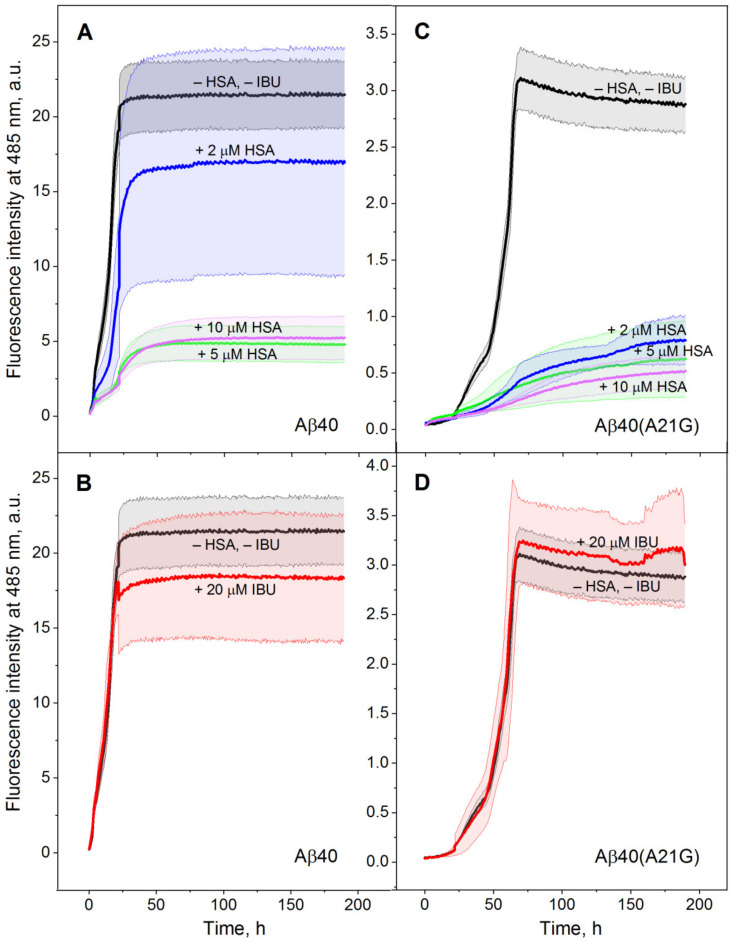
Kinetics of 20 µM Aβ_40_/Aβ_40_(A21G) fibrillation at 30 °C depending on the presence of 2–10 µM HSA (**A**,**C**) or 20 µM IBU (**B**,**D**), monitored using ThT fluorescence assay (25 mM Tris-HCl, 140 mM NaCl, 4.9 mM KCl, 1 mM MgCl_2_, 2.5 mM CaCl_2_, pH 7.4 with 0.05% NaN_3_). The standard deviations of the fluorescence signals are indicated. Excitation at 440 nm; emission at 485 nm.

**Figure 3 ijms-23-06168-f003:**
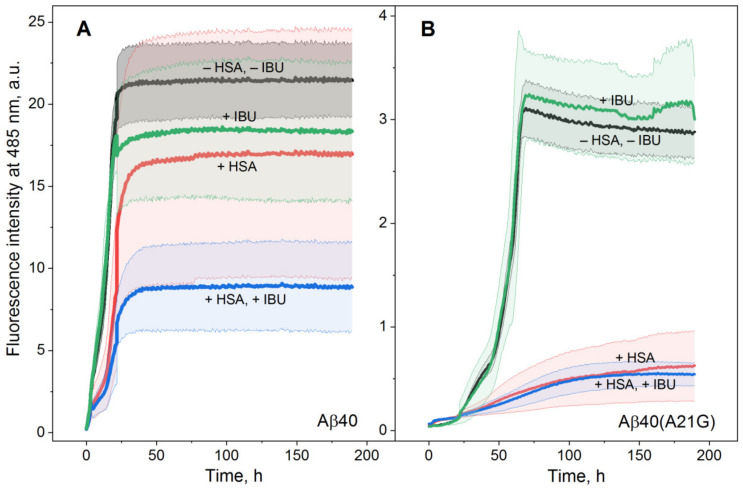
Kinetics of 20 µM Aβ_40_ (**A**)/Aβ_40_ (A21G) (**B**) fibrillation at 30 °C depending on the presence of 2 µM HSA and 20 µM IBU, monitored using ThT fluorescence assay (25 mM Tris-HCl, 140 mM NaCl, 4.9 mM KCl, 1 mM MgCl_2_, 2.5 mM CaCl_2_, pH 7.4 with 0.05% NaN_3_). The standard deviations of the fluorescence signals are indicated. Excitation at 440 nm; emission at 485 nm.

**Figure 4 ijms-23-06168-f004:**
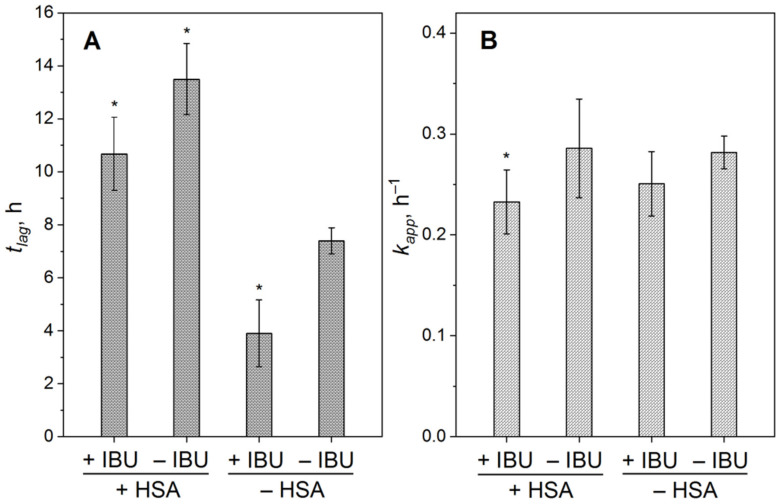
The lag time *t_lag_* (**A**) and rate constant *k_app_* (**B**) of Aβ_40_ fibrillation at 30 °C, estimated from the ThT fluorescence assay data (Figure 3A; 2 µM HSA, 20 µM IBU) using Boltzmann sigmoid function (Equation (2)). The standard deviations are shown (n = 3–5). The statistically significant differences from the experiments in the absence of HSA and IBU are indicated by * (Student’s *t*-test, *p* = 0.05).

**Figure 5 ijms-23-06168-f005:**
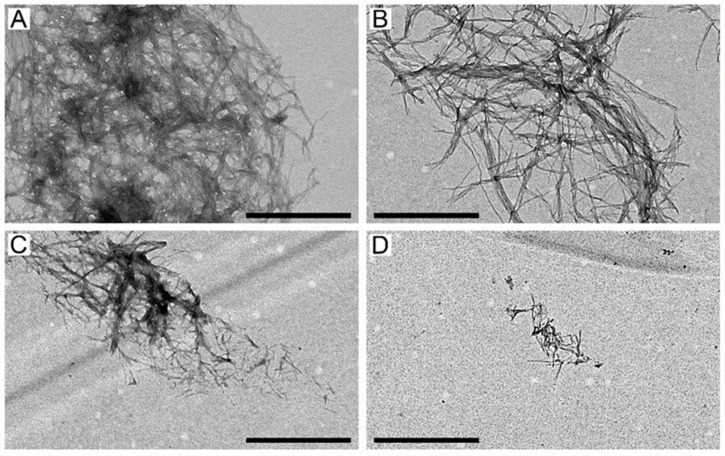
Negative staining TEM images of the 20 μM Aβ_40_ fibers grown in the course of the ThT fluorescence assay shown in Figure 3A in the absence (**A**) or in the presence of 20 μM IBU (**B**), 2 μM HSA (**C**), and 2 μM HSA and 20 μM IBU (**D**). The scale bars represent 1 μm.

**Figure 6 ijms-23-06168-f006:**
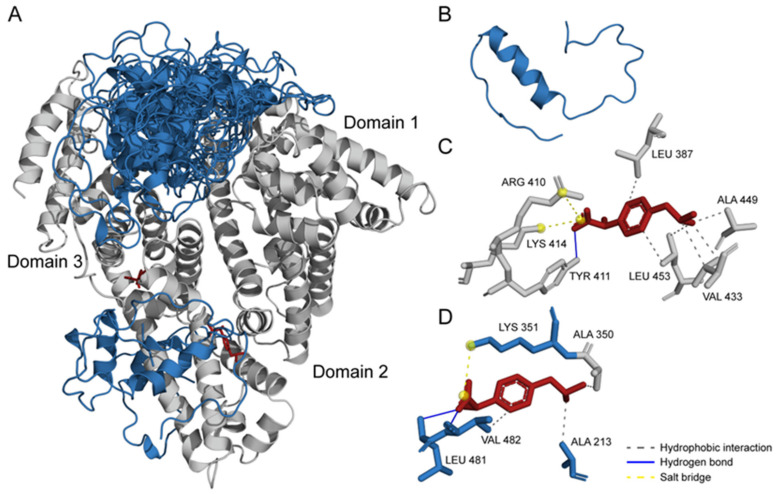
(**A**) Overlay of the representative models of the HSA–Aβ_40_ complex calculated using ClusPro docking server [54] and structure of the HSA–IBU complex (PDB entry 2BXG; HSA chain is hidden due to negligible structural changes upon IBU binding to HSA). HSA, Aβ_40_, and IBU molecules are shown in gray, blue, and red, respectively. (**B**) Tertiary structure of the folded state of Aβ_40_ (PDB entry 2LFM). (**C**,**D**) Representation of the HSA residues interacting with IBU (PDB entry 2BXG) according to the PLIP service [55]. The residues also involved in the Aβ_40_ binding (see **A**) are highlighted in blue.

**Figure 7 ijms-23-06168-f007:**
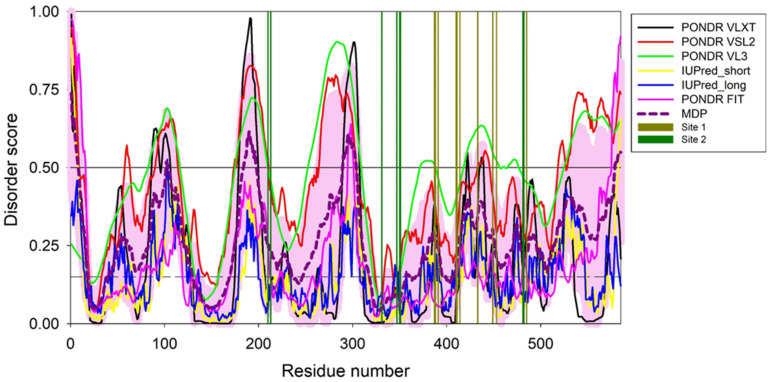
Multiparametric evaluation of the per-residue intrinsic disorder predisposition of the mature HSA (residues 25–609 of UniProt ID: P02768) evaluated by six commonly used disorder predictors: PONDR^®^ VLXT (black line), PONDR^®^ VSL2 (red line), PONDR^®^ VL3 (green line), PONDR^®^ FIT (pink line), IUPred2-Short (yellow line), and IUPred2_Long (blue line). The mean disorder calculated by averaging the outputs of the individual predictors is shown by the bold dashed dark-pink line. Light-pink shadow shows distribution of errors of mean. The disorder threshold of 0.5 (thin black line) separates residues to ordered or intrinsically disordered with the corresponding predicted disorder scores <0.5 and ≥0.5, respectively. Flexible residues are characterized by the disorder scores ranging from 0.15 (shown by thin dashed line) and 0.5. Positions of residues involved in the IBU biding sites 1 (primary) and 2 (secondary) are shown by vertical dark-yellow and dark-green bars.

**Table 1 ijms-23-06168-t001:** Parameters of the heterogeneous ligand model (1) describing the SPR data on kinetics of the HSA–Aβ interaction in the absence or presence of IBU (see Figure 1).

[IBU], µM	*k_a_*_1_ × 10^−3^, M^−1^s^−1^	*k_d_*_1_ × 10^6^, s^−1^	*K_d_*_1_ × 10^10^, M	*k_a_*_2_ × 10^−3^, M^−1^s^−1^	*k_d_*_2_ × 10^6^, s^−1^	*K_d_*_2_ × 10^10^, M
Aβ_40_						
0	65 ± 15	3.4 ± 0.6	**0.52 ± 0.09**	8.0 ± 0.8	7.9 ± 0.6	**9.9 ± 0.9**
100	35 ± 5	0.43 ± 0.05	**0.16 ± 0.02**	5.7 ± 0.6	2.2 ± 0.2	**3.9 ± 0.4**
Aβ_40_(A21G)						
0	30 ± 3	0.43 ± 0.06	**0.14 ± 0.02**	2.7 ± 0.3	1.8 ± 0.2	**6.7 ± 0.6**
100	25 ± 4	0.20 ± 0.03	**0.082 ± 0.012**	2.4 ± 0.3	1.03 ± 0.10	**4.2 ± 0.6**
			Aβ_42_			
0	50 ± 5	3.0 ± 0.5	**0.60 ± 0.06**	7.2 ± 1.1	4.9 ± 0.6	**6.9 ± 0.7**
100	40 ± 4	0.46 ± 0.06	**0.11 ± 0.02**	8.6 ± 0.8	1.4 ± 0.2	**1.7 ± 0.2**

## Data Availability

Not applicable.

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
