# Peer review of "Ibuprofen Favors Binding of Amyloid-β Peptide to Its Depot, Serum Albumin"

_ijms, 2022, doi:10.3390/ijms23116168_

Round 1
Reviewer 1 Report
The deposition of amyloid β-peptide (Aβ) in the brain is a critical event in progression of Alzheimer’s disease (AD). This Aβ deposition could be prevented by directed enhancement of Aβ binding to human serum albumin (HSA). The present study shows that ibuprofen (IBU), an exogenous HSA ligand improves HSA affinity to monomeric Aβ.
Specific comments
- The statement that an enhancement of HSA-Aβ interaction by IBU and the strengthened inhibition of Aβ fibrillation by HSA in the presence of IBU could contribute to the neuroprotective effects observed in mouse and human studies of AD does not take into account the IBU-Aβ interaction suggested by the effect of ibuprofen on the fibrillation process (figures 3 and 4). The authors are encouraged to provide experimental evidence of an IBU-Aβ interaction. SPR spectroscopy measurements will address the question elegantly. The recombinant human Aβ40/Aβ42 or Flemish variant of Aβ40 will be immobilized on the surface of the SPR sensor by amine coupling and a fresh IBU solution will be passed over the chip surface.
- Negative staining TEM images do not support the statement that 20 µM IBU alone does not induce notable suppression of the fibrillation process. Moreover what is the link between fibrils shape and the Aβ activity. Are those IBU-modified fibers still pathogenic is a key question that is not addressed/discussed.
- Efficient suppression of Aβ40/Aβ42 fibrillation was previously reported at IBU concentration as low as 10 µM [49]. To explain the discrepancy with the present work the authors suggest that it could be due to the use in the previous study of synthetic Aβ rather than human Aβ expressed in E. coli. What could be the difference between synthetic and Human Aβ expressed in E. coli.
Author Response
The deposition of amyloid β-peptide (Aβ) in the brain is a critical event in progression of Alzheimer’s disease (AD). This Aβ deposition could be prevented by directed enhancement of Aβ binding to human serum albumin (HSA). The present study shows that ibuprofen (IBU), an exogenous HSA ligand improves HSA affinity to monomeric Aβ.
Specific comments
1) The statement that an enhancement of HSA-Aβ interaction by IBU and the strengthened inhibition of Aβ fibrillation by HSA in the presence of IBU could contribute to the neuroprotective effects observed in mouse and human studies of AD does not take into account the IBU-Aβ interaction suggested by the effect of ibuprofen on the fibrillation process (figures 3 and 4). The authors are encouraged to provide experimental evidence of an IBU-Aβ interaction. SPR spectroscopy measurements will address the question elegantly. The recombinant human Aβ40/Aβ42 or Flemish variant of Aβ40 will be immobilized on the surface of the SPR sensor by amine coupling and a fresh IBU solution will be passed over the chip surface.
ANSWER:
Unfortunately, sensitivity of Bio-Rad ProteOn™ XPR36 SPR spectrometer used in our study is insufficient for detection of the direct interaction between Aβ40/Aβ42 and ibuprofen using the latter as an analyte. In fact, ibuprofen interaction with Aβ monomers, fibrils and plaques has been previously addressed in the following papers:
- Lockhart, C., S. Kim and D. K. Klimov (2012). "Explicit Solvent Molecular Dynamics Simulations of Aβ Peptide Interacting with Ibuprofen Ligands" The Journal of Physical Chemistry B 116(43): 12922-12932.
- Agdeppa, E. D., V. Kepe, A. Petri, N. Satyamurthy, J. Liu, S. C. Huang, G. W. Small, G. M. Cole and J. R. Barrio (2003). "In vitro detection of (S)-naproxen and ibuprofen binding to plaques in the Alzheimer’s brain using the positron emission tomography molecular imaging probe 2-(1-(6-[(2-(18F)fluoroethyl)(methyl)amino]-2-naphthyl)ethylidene)malononitrile" Neuroscience 117(3): 723-730.
- Hirohata, M., K. Ono, H. Naiki and M. Yamada (2005). "Non-steroidal anti-inflammatory drugs have anti-amyloidogenic effects for Alzheimer's β-amyloid fibrils in vitro" Neuropharmacology 49(7): 1088-1099.
- Raman, E. P., T. Takeda and D. K. Klimov (2009). "Molecular dynamics simulations of Ibuprofen binding to Abeta peptides" Biophys J 97(7): 2070-2079.
We have inserted these references into the manuscipt text – see section 2.2 and Conclusions chapter.
It should be mentioned that the literature data evidence prevention of Aβ fibrillation/aggregation by IBU (Hirohata et al., 2005; Agdeppa et al., 2003). Thus, the enhancement of HSA-Aβ interaction by IBU, the strengthened inhibition of Aβ fibrillation by HSA in the presence of IBU, and the direct prevention of Aβ fibrillation/aggregation by IBU synergistically prevent formation of the deleterious multimeric forms of Aβ peptide. We have inserted the respective considerations into the Conclusions chapter.
2) Negative staining TEM images do not support the statement that 20 µM IBU alone does not induce notable suppression of the fibrillation process. Moreover what is the link between fibrils shape and the Aβ activity. Are those IBU-modified fibers still pathogenic is a key question that is not addressed/discussed.
ANSWER:
We agree that in the case of Ab40 there is a tendency to suppression of its fibrillation by IBU, but this effect is statistically insignificant. We have added the corresponding remark into the text. Besides, the description of the TEM data has been extended by a more detailed comparison with the data of ThT fluorescence assay.
Aβ fibrils are considered to be less toxic compared to small Aβ oligomers (reviewed in EBioMedicine 2016, 6, 42–49; doi:10.1016/j.ebiom.2016.03.035). Nevertheless, our attempts to detect small Aβ oligomers in the prefibrillar phase using chemical crosslinking and dynamic light scattering methods showed presence of only fairly large Aβ particles with a diameter exceeding 20 nm.
It should be noted, that the main message of the manuscript, emphasized in its title, is the ability of ibuprofen to force human serum albumin to exclude Aβ monomers from the deleterious multimerization reactions more efficiently. For instance, the decline in free Aβ concentration by a factor of two would decrease Aβ multimerization rate by a factor of four, in the case of a bimolecular reaction. Thus, the paper is aimed at direct slowdown/avoidance of accumulation of the Aβ multimers.
We have added the more detailed description of the mechanisms of IBU action on the Aβ multimerization process to the Conclusions chapter.
3) Efficient suppression of Aβ40/Aβ42 fibrillation was previously reported at IBU concentration as low as 10 µM [49]. To explain the discrepancy with the present work the authors suggest that it could be due to the use in the previous study of synthetic Aβ rather than human Aβ expressed in E. coli. What could be the difference between synthetic and Human Aβ expressed in E. coli.
ANSWER:
The differences between properties of the recombinant and synthetic Ab forms are described in the following paper:
Finder, V. H., I. Vodopivec, et al. (2010). "The Recombinant Amyloid-β Peptide Aβ1–42 Aggregates Faster and Is More Neurotoxic than Synthetic Aβ1–42" Journal of Molecular Biology 396(1): 9-18.
The recombinant Aβ formed fibrils substantially faster than synthetic Aβ sample due to a variety of impurities in synthetic Aβ preparation, including peptides with D-histidine. Besides, the recombinant Aβ was significantly more neurotoxic both under in vitro and in vivo conditions. We have included the respective explanation into the manuscript text.
Reviewer 2 Report
The manuscript by Litus et al. presents experimental and simulation study of ibuprofen influence to an interaction of human serum albumin with the amyloid peptide associated with Alzheimer’s decease. Based on their previous works and current results, the authors suggested a novel mechanism of inhibition of Aβ fibrillation by albumin in the presence of ibuprofen. Thus, this solid paper is likely to be of interest to the Journal readership.
Minor changes I would suggest prior to publication are:
- a) the manuscript does not discuss the effect of ibuprofen on the formation of small Aβ oligomers, which are more toxic to neurons than fibrils;
- b) what is the putative molecular mechanism (e.g. revealed by modeling) for the A21G AD mutation to increase the affinity of HSA for Aβ40?
Author Response
The manuscript by Litus et al. presents experimental and simulation study of ibuprofen influence to an interaction of human serum albumin with the amyloid peptide associated with Alzheimer’s decease. Based on their previous works and current results, the authors suggested a novel mechanism of inhibition of Aβ fibrillation by albumin in the presence of ibuprofen. Thus, this solid paper is likely to be of interest to the Journal readership.
Minor changes I would suggest prior to publication are:
a) the manuscript does not discuss the effect of ibuprofen on the formation of small Aβ oligomers, which are more toxic to neurons than fibrils;
ANSWER: We agree that such analysis would be of interest, but our attempts to detect small Aβ oligomers in the prefibrillar phase using chemical crosslinking and dynamic light scattering methods showed presence of only fairly large Aβ particles with a diameter exceeding 20 nm.
It should be noted, that the main message of the manuscript, emphasized in its title, is the ability of ibuprofen to force human serum albumin to exclude Aβ monomers from the deleterious multimerization reactions more efficiently. For instance, the decline in free Aβ concentration by a factor of two would decrease Aβ multimerization rate by a factor of four, in the case of a bimolecular reaction. Thus, the paper is aimed at direct slowdown/avoidance of accumulation of the Aβ multimers.
We have added the more detailed description of the mechanisms of IBU action on the Aβ multimerization process to the Conclusions chapter.
b) what is the putative molecular mechanism (e.g. revealed by modeling) for the A21G AD mutation to increase the affinity of HSA for Aβ40?
ANSWER: Since residue A21 of Aβ40 molecule directly interacts with HSA in 7 of the 25 model structures of HSA-Aβ40 complex (Figure 6), the A21G substitution is expected to affect the HSA-Aβ40 interaction, but direction of the effect is barely predictable. We have added this consideration into the manuscript text.
Round 2
Reviewer 1 Report
The authors have made an appreciate effort to address the questions raised and most points have been addressed satisfactorily and adequately.